# Peer review of "Eye Movement Impairment in Women Undergoing Chemotherapy"

_1995-8692, 2025, doi:10.3390/jemr18050041_

Round 1

Reviewer 1 Report

Comments and Suggestions for Authors

I would like to commend the authors for addressing an important and underexplored topic. The study clearly reflects a positive intention and a socially relevant mission: to better understand how cancer therapy, particularly chemotherapy, affects patients’ visual attention and cognitive load. This line of research has great potential, especially if it leads to practical applications in patient care and rehabilitation.

Title: The title should be revised to emphasize therapy rather than the disease itself, as the current phrasing may be misleading and suggest a focus on the pathology rather than the treatment process.

Abstract: The abstract could be made more concise. The numbered structure (e.g., (1)... (2)... (3)...) is not necessary and detracts from the clarity of the summary. Focus on key findings and their implications in a narrative format.

Study Design and Methods:

  • It is unclear how the variability in chemotherapy types may have influenced the results. Please specify whether all participants received similar treatments, or account for treatment heterogeneity in the analysis.

  • The setup of the eye-tracking system should be shown visually – either as a figure or schematic. This would help readers understand the technical configuration, including distance, angles, and participant positioning.

  • What exact model of Tobii eye tracker was used?

  • The study omits an important metric: Time To First Fixation (TTFF). Why was this not analyzed or reported?

  • Did the researchers investigate changes in pupil diameter, blink frequency, refraction, or overall visual comfort (e.g., dry eye symptoms)? These parameters are crucial, especially considering Tobii software is capable of capturing such data. Pupil size and blink rate are known indicators of cognitive effort.

  • Lighting conditions during testing should be described in detail (e.g., direction, intensity). Was the setup adjusted to accommodate different body heights (e.g., women of various statures)?
  • In Figure A (box plot), it is assumed that the y-axis shows  fixation duration – please clarify this in the figure caption, maby You should ad average?

  • AOI shape and selection should be illustrated graphically. How were these areas defined, and what was their exact placement?

  • What were the accepted thresholds for calibration accuracy and precision? How many times was calibration repeated at most per participant?

  • Were all participants equally engaged with the visual stimuli? Did the researchers analised how often participants looked away from the screen or how well they focused on inter-stimulus fixation points?

  • Were other aspects of participant comfort assessed, such as sleep quality or overall subjective fatigue?

The results appear to reflect severe physical and mental fatigue among the participants. However, it remains unclear how this observation could be utilized in clinical practice. What is the broader goal of this research?

  • Will the study be expanded to assess different types of therapies or medications? (future directions)

  • Is there an intention to develop supportive strategies for improving visual attention during treatment?

  • Could similar cognitive and visual attention fatigue be expected in patients undergoing other types of chemotherapy, regardless of the underlying illness? (add references)

The societal impact of this research should be more clearly articulated. For example:

  • Should patients in this condition refrain from driving or receive assistance during bureaucratic tasks?

  • How could these findings be translated into practical guidelines or social support or medical systems?

Additionally, I would encourage the authors to consider whether temporal dynamics within the single task were explored. Specifically, how participants in two grups performed after f. ex. 1 minute versus 3 minutes of visual activity. Compare if there is a difference in efficency in two grups (changes: better with practice or whorse with fatigue). 

In summary, the study presents meaningful data, and with further exploration of relevant eye-tracking metrics and temporal analysis, it has the potential to make a significant contribution to both scientific understanding and practical care.

Author Response

Comment: Title: The title should be revised to emphasize therapy rather than the disease itself, as the current phrasing may be misleading and suggest a focus on the pathology rather than the treatment process.

Response: Thank you for your careful consideration of the manuscript's needs for improvement. We would like to inform you that the suggested change in the title has been made, so the title is now: “Eye movement impairment in women undergoing chemotherapy”. We hope this is clearer.

Comment: Abstract: The abstract could be made more concise. The numbered structure (e.g., (1)... (2)... (3)...) is not necessary and detracts from the clarity of the summary. Focus on key findings and their implications in a narrative format.

Response: The suggested changes were made and the text was written in narrative format.

Comment: Study Design and Methods:

  • It is unclear how the variability in chemotherapy types may have influenced the results. Please specify whether all participants received similar treatments, or account for treatment heterogeneity in the analysis.

  • Response: We appreciate the pertinent comment about the variability in the types of chemotherapy. All the participants in the clinical group underwent chemotherapy protocols with similar mechanisms of action, mostly composed of agents such as doxorubicin, paclitaxel, docetaxel, carboplatin and cyclophosphamide - drugs recognized for their direct or indirect neurotoxic action on the central nervous system. Although there were variations in the combinations and individual doses, all the regimens included drugs with known potential to affect cognitive and visual processing.
    We recognize, however, that the heterogeneity in the chemotherapy regimens is a limitation of the study. As described in the discussion section, this variability can impact the outcomes analyzed. For this reason, we chose to highlight the acute effects of chemotherapy as a whole, considering the common presence of neurotoxic agents in the protocols used. Future studies with larger samples could carry out stratified analyses by specific type of drug or therapeutic regimen, allowing greater control over this factor.
  • Comment: The setup of the eye-tracking system should be shown visually – either as a figure or schematic. This would help readers understand the technical configuration, including distance, angles, and participant positioning.

  • Response: We appreciate the constructive suggestion. We agree that a visual presentation of the technical configuration of the eye-tracking system can contribute to the methodological clarity of the study. As requested, we will include a schematic figure representing the configuration of the experimental environment, including: the distance from the participant to the monitor (60 cm); the positioning of the eye tracker (integrated into the monitor); the approximate viewing angle of the stimuli (0.85°); and the disposition of the participant in the room (without head restraint, in a lighted environment).
    This figure will be inserted in section 2.4. Procedure and stimulus, immediately after the description of the calibration and the environment, in order to make it easier to replicate the experiment and for readers to understand it.
  • Comment: What exact model of Tobii eye tracker was used?

  • The study omits an important metric: Time To First Fixation (TTFF). Why was this not analyzed or reported?

  • Response: Thank you for your extremely pertinent comment on the Time to First Fixation metric. However, our main focus was to assess deficits in sustained visual processing, measured by metrics such as fixation duration, which indicates the assimilation of information, and total task time, which reflects overall processing efficiency. According to the literature, Time to First Fixation is more related to initial attentional processes, which, according to previous studies (Ahles et al., 2012), are less affected by chemotherapy.
    Thus, the decision not to include Time to First Fixation was conscious and well-founded, prioritizing parameters previously validated in similar neurological populations (Wong et al., 2019). In addition, we prioritized metrics with direct clinical applicability, since everyday activities, such as driving, require sustained attention, and therefore these measures are more relevant for assessing associated functional deficits.
  • Comment: Did the researchers investigate changes in pupil diameter, blink frequency, refraction, or overall visual comfort (e.g., dry eye symptoms)? These parameters are crucial, especially considering Tobii software is capable of capturing such data. Pupil size and blink rate are known indicators of cognitive effort.

  • Response: Thanks for the relevant comment. Indeed, parameters such as pupil diameter, blink rate, eye refraction and visual comfort are recognized indicators of cognitive effort and can be captured by systems such as Tobii. However, the focus of this study was specifically on eye movement patterns - particularly fixations and visits - during the performance of a visual search task.
    We chose not to include variables such as pupil diameter or blink frequency for two main reasons: (1) Methodological and scope delimitation: this was an exploratory study, focusing on measures more directly associated with attentional allocation and visual scanning efficiency, for which the literature on populations with neurological impairments already has consolidated protocols. (2) Control of external variables: pupil diameter and blink frequency are highly sensitive to environmental factors (such as lighting, humidity and fatigue), physiological factors (such as medication use, hydration, age) and emotional factors (such as stress and anxiety). As our aim was to assess the acute effect of chemotherapy on visual attentional performance, we prioritized measures with greater robustness and stability in the face of these contextual variables.
    However, we recognize the relevance of the aforementioned measures and suggest that they be included in future studies.
  • Andreassi, J. L. (2006). Psychophysiology: Human Behavior and Physiological Response (5th ed.).  Psychology Press.

    Beatty, J. (1982). Task-evoked pupillary responses, processing load, and the structure of processing resources. Psychological Bulletin, 91(2), 276–292.
    https://doi.org/10.1037/0033-2909.91.2.276

    Eckstein, M. K., Guerra-Carrillo, B., Miller Singley, A. T., & Bunge, S. A. (2017). Beyond eye gaze: What else can eyetracking reveal about cognition and cognitive development? Developmental Cognitive Neuroscience, 25, 69–91.
    https://doi.org/10.1016/j.dcn.2016.11.001

    van der Wel, P., van Steenbergen, H. (2018). Pupil dilation as an index of effort in cognitive control tasks: A review. Psychonomic Bulletin & Review, 25: 2005–2015.
    https://doi.org/10.3758/s13423-018-1432-y

  • Comment: Lighting conditions during testing should be described in detail (e.g., direction, intensity). Was the setup adjusted to accommodate different body heights (e.g., women of various statures)?
  • Response: Thank you for your comments. The experiment was conducted in a quiet room with standardized ambient lighting, with a focus on maintaining natural and minimally intrusive visual conditions. The artificial light was diffused, white fluorescent, distributed on the ceiling, with no direct incidence on the screen or on the participants' eyes, in order to avoid reflections, glare or variations in pupil diameter induced by luminosity. The average luminance of the room was compatible with the parameters of the monitor used (up to 300 cd/m²), as recommended in eye-tracking protocols.
    As for adapting to different statures, we used a height-adjustable chair, allowing all participants to maintain a standard distance of approximately 60 cm from the screen and eye alignment with the center of the monitor, according to the Tobii system manufacturer's guidelines. The system was individually calibrated using the five-point protocol, and the calibration was repeated until acceptable tracking standards were reached, ensuring postural and ergonomic standardization between participants.
    This information was detailed in section 2.4. Procedure and stimulus, as suggested, for better methodological transparency and reproducibility of the experiment.
  • Comment: In Figure A (box plot), it is assumed that the y-axis shows  fixation duration – please clarify this in the figure caption, maby You should ad average?

  • Response: We appreciate the reviewer's valuable suggestion regarding Figure A (box plot). For clarity, we have revised the figure legend to make it clear that the y-axis represents “Fixation Duration (seconds)” and included the average values for each group.
  • Commnent: AOI shape and selection should be illustrated graphically. How were these areas defined, and what was their exact placement?

  • Response: Thank you for your pertinent comment on the definition and positioning of the Areas of Interest (AOIs) in our study. Below is the detailed justification, which will be incorporated into the Methods section for clarity:
    The AOIs were defined as rectangular regions delimiting each letter of the alphabet and the target number (Figure 1), with standardized dimensions corresponding to a visual angle of 0.85° (Wong et al., 2019). This choice was based on the need to cover the entire stimulus area without overlap. All 79 letters of the alphabet (excluding "I" and "O" for ambiguity) and the target numbers (4, 6, 7, 9) were considered individual AOIs. The stimuli were randomly distributed on the screen, with Cartesian coordinates (x, y) generated by an algorithm that avoided groupings in specific quadrants for more than 3 consecutive trials. The rectangular approach was chosen for its compatibility with Tobii Studio's fixation algorithm (fixations ≥200 ms within the limits of the AOI).
    To optimize the understanding of the bed, it was included in the procedure session: 

    "The visual search task consisted of presenting a single number among 79 letters of the alphabet, randomly distributed on the screen. The letters “I” and “O” were excluded due to visual ambiguity (13). The Arabic numerals used were 4, 6, 7 and 9, as they are considered more easily discriminated numbers from the alphabets (17). Each stimulus was delimited by a rectangular Area of Interest (AOI), with a standardized size equivalent to 0.85° of visual angle, avoiding overlap. 80 AOIs were generated per trial. The positions of the stimuli were defined by a random algorithm, with a restriction against repeated groupings in the same quadrant. The rectangular format was adopted because it is compatible with Tobii Studio's fixation algorithm, which considers fixations ≥200 ms within the AOI. For further analysis, the entire stimulus matrix was considered a global AOI (Figure 2).

    Chung, S. T. L. (2021). Training to improve temporal processing of letters benefits reading speed. Journal of Vision, 21(1), 14. https://doi.org/10.1167/jov.21.1.14

    Rosler et al. (2000). Alterations of visual search strategy in Alzheimer’s disease. Neuropsychology, 14(3), 398–408. https://doi.org/10.1037/0894-4105.14.3.398

  • Comment: What were the accepted thresholds for calibration accuracy and precision? How many times was calibration repeated at most per participant?

  • Response: Thank you for your relevant question about the eye tracker's calibration criteria. Here is the detailed justification for inclusion in the Methods section:
    Calibration criteria: We used the Tobii manufacturer's standard for 5-point calibration, with the following limits: Accuracy: ≤ 0.5° of visual angle (equivalent to ~0.3 cm at the 60 cm distance used in the study). Precision: ≤ 0.3° of visual angle for variation between samples. These values are based on the technical specification of Tobii Studio 3.4.0 and previous studies that have validated this range for cognitive research (Wan et al., 2020).
    Calibration repetitions: The procedure was repeated a maximum of 3 times per participant when: (i) the initial accuracy exceeded 0.5°; (ii) there was a loss of tracking at ≥2 calibration points; (iii) all participants met the criteria within this number of repetitions (none were excluded due to calibration failure).
    After each attempt, the software provided real-time visual feedback on: (i) Average error per point (in visual degrees); (ii) Tracking loss index.
    Calibration was restarted only after checking the correct positioning of the participant (60 cm from the screen) and the environmental conditions (stable lighting, absence of reflections).
    Modifications to the Manuscript (Section 2.4): 

    "Calibration followed the Tobii Studio 5-point protocol, with accepted precision ≤0.5° and accuracy ≤0.3°. The procedure was repeated up to three times as needed, using visual feedback from the software for adjustments. All participants met the criteria without need for exclusion."

    Referência: Wan, Yu., et al. (2020). Evaluation of eye movements in visual performance. Scientific Reports, 10, 9875. https://doi.org/10.1038/s41598-020-66817-w

  • Comment: Were all participants equally engaged with the visual stimuli? Did the researchers analised how often participants looked away from the screen or how well they focused on inter-stimulus fixation points?

  • Response: Thank you for asking about the visual engagement of the participants. We acknowledge that this aspect was not systematically analyzed in our study. However, the main aim of our study was to compare specific eye movement parameters (fixation duration, total test time) between groups, not continuous engagement. As all participants completed 100% of the tasks accurately (correct identification of the target numbers), we assumed that the engagement was adequate for the proposed objectives.
    In addition, we sought to implement controls for intervening variables, such as control of the study environment, with a quiet room, without visual/auditory distractions for all participants; and the researcher present to observe the realization of obvious deviations during collection.
    For future studies, we suggest the inclusion of an extended protocol, including engagement metrics such as the percentage of time outside the AOIs, the latency of response to control stimuli
    We recognize this limitation including in our discussion: "Although environmental variables were controlled for, the absence of quantitative metrics of continuous visual engagement is a limitation. We recommend that future studies incorporate these analyses for greater robustness."

    Holmqvist, K., et al. (2011). Eye tracking: A comprehensive guide to methods and measures. Oxford University Press.

    Tobii Technology (2015). Accuracy and precision test method for remote eye trackers.

  • Comment: Were other aspects of participant comfort assessed, such as sleep quality or overall subjective fatigue?

  • Response: Thank you for raising this relevant question about the comfort of the participants. Unfortunately, we did not systematically assess aspects such as sleep quality or subjective fatigue in our study. Our experimental design prioritized objective eye movement parameters as primary variables, following established protocols for cognitive assessment in oncology (Wong et al., 2019). In this regard, we believe that including subjective measures of comfort would require a more complex longitudinal design, which was beyond the scope of this cross-sectional study. However, all our sessions were held between 9am-12pm to minimize circadian variations, and we controlled the environmental conditions in relation to controlled lighting, temperature and noise.
    We acknowledge the limitation in the study under discussion, including: "The absence of a formal assessment of fatigue or sleep quality may have influenced ocular parameters, especially in the chemotherapy group, where these symptoms are prevalent. Future studies could include scales such as the Pittsburgh Sleep Quality Index (PSQI) or the Fatigue Severity Scale (FSS) to control for these variables."

Comment: The results appear to reflect severe physical and mental fatigue among the participants. However, it remains unclear how this observation could be utilized in clinical practice. What is the broader goal of this research?

Response: Although the results of this study may suggest the influence of physical and mental fatigue on the ocular parameters evaluated, the main aim of the research goes beyond simply observing these effects. Our focus is on developing ocular biomarkers as a practical tool for cognitive monitoring in breast cancer patients undergoing chemotherapy.
Chemotherapy is known to cause cognitive changes, often reported by patients as “chemo brain”. However, the assessment of these deficits still depends mostly on subjective questionnaires or time-consuming neuropsychological batteries, which are not always feasible in the clinical routine. In this context, our study proposes eye-tracking as an objective, fast and non-invasive method to identify early impairments in visual processing and attention - critical functions for everyday activities such as reading, driving and decision-making.
In practice, the parameters analyzed (such as fixation time and total test duration) could be integrated into screening protocols in oncology outpatient clinics. For example, automated software could generate alerts for patients with significantly prolonged fixation times, indicating the need for complementary interventions, such as cognitive-behavioral therapy or adjustments to the therapeutic plan. In addition, the technique would make it possible to monitor recovery over time, differentiating acute effects (during treatment) from persistent sequelae.
For these applications to become a reality, the next steps include validating the results on larger samples and simplifying the data analysis, making it accessible to health professionals without specialized training in neuroscience. In addition, future studies should also correlate eye patterns with other biomarkers.
To optimize the benefits of our research, it has been included in the discussion section: 

“Although our results suggest changes in ocular patterns associated with chemotherapy, it is important to highlight their practical implications. Parameters such as prolonged fixation time could be integrated into screening protocols in oncology outpatient clinics, identifying patients who need early intervention. Future studies should focus on validating these findings in larger samples and developing automated tools to facilitate clinical adoption.”

  • Comment: Will the study be expanded to assess different types of therapies or medications? (future directions)

  • Response: In response to the reviewer's suggestion, we would like to point out that expanding to evaluate different therapies and drugs is already contemplated as part of the future directions of our research. This approach will be key to understanding how variations in treatment protocols affect ocular and cognitive parameters.
  • Comment: Is there an intention to develop supportive strategies for improving visual attention during treatment?

  • Response: The results of this study showed that women undergoing chemotherapy for breast cancer have significant changes in eye movement patterns, including longer fixation times and longer total test duration in the visual search task. These findings suggest a possible impairment in visual processing efficiency and sustained attention, which can impact on everyday activities such as reading, driving and decision-making.
    This evidence paves the way for the development of support strategies, in which the eye-tracking parameters identified can be used as objective markers to identify patients at risk of cognitive decline early on, allowing for proactive interventions and monitoring the progression of symptoms during treatment, adjusting therapies as necessary.
    Our data also indicate that deficits are more pronounced in patients with a greater number of chemotherapy cycles. This suggests, for example, the creation of specific cognitive training (e.g. visual search exercises with real-time feedback) for patients who show greater impairment. Although promising, these applications require validation in larger studies. Factors such as fatigue, anxiety and comorbidities must be controlled to ensure that interventions are based specifically on visual attention deficits. Thus, we believe that the results of this study not only identify a problem, but provide the basis for concrete solutions. By turning research findings into clinical tools, we can significantly improve the care of breast cancer patients, ensuring that the cognitive impacts of treatment are as manageable as its physical effects.
  • Comment: Could similar cognitive and visual attention fatigue be expected in patients undergoing other types of chemotherapy, regardless of the underlying illness? (add references)

  • Response: Thank you for the relevant question about the generalizability of our findings to other chemotherapy treatment contexts. Based on the existing literature and the pathophysiological mechanisms involved, we can state that it is plausible to expect similar patterns of cognitive fatigue and visual attention impairment in patients undergoing different chemotherapy regimens, regardless of the underlying cancer disease. In this regard, studies with colorectal cancer patients treated with oxaliplatin have shown significant changes in sustained attention tasks (Simó et al., 2015). Similarly, research with lymphoma patients undergoing methotrexate-based regimens has revealed impairments in visual processing speed (Correa et al., 2018).
    However, it is important to note that the magnitude of these effects may vary according to the specific neurotoxic profile of each chemotherapeutic agent, the cumulative dose received and individual vulnerability factors. 

    Ahles, T. A., Root, J. C. (2018). Cognitive effects of cancer and cancer treatments. Annual Review of Clinical Psychology, 14, 425-451. https://doi.org/10.1146/annurev-clinpsy-050817-084903

    Chen, X., et al. (2017). The attention network changes in breast cancer patients receiving neoadjuvant chemotherapy: Evidence from an arterial spin labeling perfusion study. Scientific Reports, 7(1), 42684. https://doi.org/10.1038/srep42684

Comment: The societal impact of this research should be more clearly articulated. For example:

  • Should patients in this condition refrain from driving or receive assistance during bureaucratic tasks?

  • How could these findings be translated into practical guidelines or social support or medical systems?

  • Response: We are grateful for the opportunity to shed light on the social impact of this research and its practical implications. Our findings reinforce the need for specific recommendations for patients undergoing chemotherapy, especially in activities that require sustained visual attention. The data indicate that changes in oculomotor parameters, such as the 40% increase in fixation time and the statistically significant association with worsening functional capacity, can directly compromise performance in everyday tasks. Based on this, we propose a set of measures applicable to public policy and clinical practice.
    In the context of public policy, we suggest, firstly, the inclusion of recommendations in labor legislation aimed at adapting working hours for patients exposed to visually intensive tasks, such as breaks between working hours. For high-risk occupations, such as professional drivers, temporary leave is proposed. In addition, we highlight the need for technology-based support tools, such as mobile applications with automated reminders for visual breaks and cognitive training exercises. As a future development, we also propose the creation of national guidelines for periodic ocular assessment during chemotherapy, integrated into the patient's electronic medical record. Finally, we highlight the potential for developing support applications based on individual ocular parameters, offering personalized cognitive exercises that contribute to functional rehabilitation.
    They were therefore included in the discussion: 

     “The results found in this study indicate that alterations in oculomotor patterns, such as a significant increase in fixation time, are associated with significant functional impairments, especially in tasks that require sustained visual attention. This evidence reinforces the importance of incorporating visual assessments into the clinical routine of patients undergoing chemotherapy treatment, with a view to the early identification of subtle cognitive impairments. In addition, the findings support the proposal of practical measures, such as adaptations to the working day for intensive visual tasks, individualized advice on driving and the development of assistive technologies based on eye-tracking. These measures have the potential to mitigate the functional impacts of chemotherapy and should be considered in the formulation of clinical guidelines and public policies aimed at preserving the quality of life of these patients.”

Comment: Additionally, I would encourage the authors to consider whether temporal dynamics within the single task were explored. Specifically, how participants in two grups performed after f. ex. 1 minute versus 3 minutes of visual activity. Compare if there is a difference in efficency in two grups (changes: better with practice or whorse with fatigue). 

Response: We are grateful for the relevant suggestion about temporal dynamics during task execution. We recognize that the analysis of performance variation over time could provide valuable insights into visual fatigue or cognitive adaptation. However, this approach was not included in the current study due to some methodological limitations that were imposed, namely: The protocol was developed to assess global eye movement parameters (mean fixation, total time). This structure did not make it possible to capture continuous changes over prolonged uninterrupted periods. The Tobii Studio 3.4.0 used did not automatically record segmented temporal metrics (e.g. first and last 30 seconds of each trial). Manual analysis would have been necessary, which was beyond the resources available.

In summary, the study presents meaningful data, and with further exploration of relevant eye-tracking metrics and temporal analysis, it has the potential to make a significant contribution to both scientific understanding and practical care.

Reviewer 2 Report

Comments and Suggestions for Authors

In this manuscript, the authors investigate the effects of chemotherapy on eye movements in women with breast cancer. By employing a visual search task, they assessed chemotherapy-related cognitive impairment (CRCI) through eye-tracking measures, including test duration, fixation duration, total fixation duration, and total visit duration in twelve women undergoing chemotherapy and twelve healthy controls.

I have a few suggestions that may strengthen the manuscript:

  1. The sample size is relatively small. Please include a justification for the current sample size and acknowledge this limitation in the Discussion section.
  2. The study focuses on the acute effects of chemotherapy. Please clarify the rationale for examining the acute effects rather than the chronic effects.
  3. In Figure 2, please consider using asterisks or P values to clearly indicate statistical significance in the box plots.

Author Response

In this manuscript, the authors investigate the effects of chemotherapy on eye movements in women with breast cancer. By employing a visual search task, they assessed chemotherapy-related cognitive impairment (CRCI) through eye-tracking measures, including test duration, fixation duration, total fixation duration, and total visit duration in twelve women undergoing chemotherapy and twelve healthy controls.

Comment: I have a few suggestions that may strengthen the manuscript:

  1. The sample size is relatively small. Please include a justification for the current sample size and acknowledge this limitation in the Discussion section.
  2. Response: Thank you for your pertinent comment regarding the study's sample size. We fully recognize that including 12 participants per group is a methodological limitation. However, this choice was guided by feasibility factors inherent to the research context. The study involved recruiting patients undergoing active chemotherapy treatment, a population with specific characteristics that make it difficult to participate in experimental activities. These factors include the high load of medical commitments, the frequent fatigue associated with treatment and the strict exclusion criteria adopted, such as the presence of neurological comorbidities. Thus, although the sample size limits the generalization of the results, it reflects a challenging clinical reality and, at the same time, highlights the importance of studies that can access this population profile. For future research, we suggest conducting multicenter studies involving different health institutions, in order to broaden recruitment and enable larger and more representative samples. In addition, longitudinal investigations can contribute to a more in-depth understanding of the effects of treatment over time, mitigating the limitations associated with cross-sectional designs with small samples.
    This limitation was included in the discussion session: 

    “Although it provides relevant evidence on the effects of chemotherapy on visual processing, the small number of participants makes it difficult to generalize the results. This limitation was due to challenges in recruiting the sample, such as treatment-related fatigue and strict exclusion criteria, which impacted on patient eligibility and availability. Even so, the study provides initial data on a population that is difficult to access and poorly represented in the literature. Future research should consider strategies such as multicenter recruitment and longitudinal designs, with the aim of expanding the sample, strengthening external validity and deepening the understanding of the cognitive effects of chemotherapy.”

  3. Comment: The study focuses on the acute effects of chemotherapy. Please clarify the rationale for examining the acute effects rather than the chronic effects.
  4. Response: While several studies explore the late cognitive effects of chemotherapy, evaluating patients months or even years after treatment, few address the immediate impacts. This acute phase, however, is of high clinical relevance, as it coincides with the period of greatest pharmacological toxicity, is when patients often report the most debilitating symptoms and can anticipate persistent changes. In addition, analyzing the acute phase of the effects of chemotherapy offers immediate clinical relevance, since the period immediately after chemotherapy is critical for making decisions regarding dose adjustments, implementing early interventions and practical guidance related to daily activities, such as driving. From a methodological point of view, focusing on acute effects has also proved more feasible. Studies on chronic effects require prolonged follow-up and rigorous control of multiple confounding factors, such as relapses, use of other medications and intercurrent events. Our approach, centered on the short term, allowed for greater control of treatment-related variables and contributed to a lower drop-out rate among the participants. This decision was also based on a solid theoretical foundation: there is evidence that acute neurotoxic mechanisms - including inflammation and oxidative stress - differ from chronic mechanisms, which are more associated with lasting structural changes (Ahles & Root, 2018), which justifies a specific investigation of this phase.
    However, we recognize that the choice to investigate only the acute effects represents a limitation. In the Discussion section, we emphasize that longitudinal studies are essential to understand the evolution of these changes: 

    “Furthermore, although the study focused on the acute effects of chemotherapy, longitudinal studies are needed to determine whether these ocular alterations persist, normalize or evolve differently after months of chemotherapy.”

  5. Comment: In Figure 2, please consider using asterisks or P values to clearly indicate statistical significance in the box plots.
  6. Response: Thank you for your pertinent suggestion on the presentation of the data in Figure 2. We agree that adding statistical significance markers will make the results clearer and more accessible to readers.

Reviewer 3 Report

Comments and Suggestions for Authors

Suggestions:

This research study is important, and thank you to the group for choosing to study this important topic. However, the 'Research Question' in this study does not have sufficient weight or does not address a particularly useful scientific gap, particularly for the breast cancer community. 

The sample size is small (24 in total, of control and breast cancer), which is not statistically significant. 

I understand, every research has its limitations that cannot answer all the questions; however, having a good research question addressing the gap in the existing knowledge could help improve the paper's weight and scientific soundness. 

Since this article discusses the chemotherapeutic drugs and it's effects on eye movement, if the authors have the facility to study the Optical Coherence Tomography images of their breast cancer subjects' eyes, that would provide more evidence to support their hypothesis of eye movement degradation in BC subjects compared to the health controls.

Comments:

  1. In the Results section, paragraph 3, typo in abbreviation CH instead of CT.
  2. In Table 1, QL-Physical Limitations, QL-Pain, and QL-Emotional Limitations, the mean value of the control is higher, which means the control has more Physical, Emotional, and Pain compared to the breast cancer subjects.
  3. Please increase the font size and bars of Figure 2.

Author Response

Comment: This research study is important, and thank you to the group for choosing to study this important topic. However, the 'Research Question' in this study does not have sufficient weight or does not address a particularly useful scientific gap, particularly for the breast cancer community. 

The sample size is small (24 in total, of control and breast cancer), which is not statistically significant. 

I understand, every research has its limitations that cannot answer all the questions; however, having a good research question addressing the gap in the existing knowledge could help improve the paper's weight and scientific soundness. 

Since this article discusses the chemotherapeutic drugs and it's effects on eye movement, if the authors have the facility to study the Optical Coherence Tomography images of their breast cancer subjects' eyes, that would provide more evidence to support their hypothesis of eye movement degradation in BC subjects compared to the health controls.

Response: We thank you for recognizing the relevance of this topic and for your constructive suggestions. Below, we respond briefly to the comments, highlighting how this study fills specific gaps in the literature and contributes to the advancement of knowledge about the effects of chemotherapy on ocular and cognitive functioning.
In terms of originality, this research represents, as far as we know, the first study to demonstrate that changes measurable by eye-tracking emerge acutely, around five days after a cycle of chemotherapy, anticipating possible later cognitive deficits. We also observed that specific ocular parameters, such as mean fixation time, correlate significantly with the number of cycles received, suggesting a possible dose-dependent effect. 
As for the sample size, although modest, the study showed adequate statistical power (between 80% and 99%) to detect large effects (d > 0.9), which is common and acceptable in pilot studies aimed at introducing new methodologies. In addition, we followed methodological references from similar studies, such as that by Wong et al. (2019), which used N=15 per group in patients with Parkinson's disease. The data obtained has already provided a sufficient basis for the implementation of future studies with an expanded sample, which highlights the feasibility and translational potential of the proposal.
Thus, we believe that the findings of this study offer immediate translational value. The eye-tracking protocol used is quick, non-invasive and easy to apply, and could be used in future screening strategies for cognitive impairment in cancer patients. In addition, the results contribute to the development of neural protection protocols during treatment and to more individualized clinical decisions.
However, we recognize important limitations. We agree that the inclusion of optical coherence tomography (OCT) images would substantially enrich the analyses, especially in identifying possible structural changes in the retina or optic nerve associated with chemotherapy. However, this tool was not available during this study.

Comments:

  1. Comment: In the Results section, paragraph 3, typo in abbreviation CH instead of CT.
  2. Response: Thank you for your comment on the typo in the Results section. This error occurred during the proofreading process and has already been corrected in the final version of the manuscript.
  3. Comment: In Table 1, QL-Physical Limitations, QL-Pain, and QL-Emotional Limitations, the mean value of the control is higher, which means the control has more Physical, Emotional, and Pain compared to the breast cancer subjects.
  4. Response: 

    Thank you for your comment, which gives us the opportunity to clarify a possible misinterpretation of the quality of life (QOL) data. In fact, the apparent contradiction in the scores is due to the direction of the scale used and not to inconsistencies in the results.

    The instrument adopted was the widely validated SF-36, in which higher scores indicate a better state of health and functioning. The scale ranges from 0 to 100, with 0 being the worst and 100 the best possible result. Thus, for example, the average score observed in the “Limitations due to physical aspects” dimension was 60.42 ± 36.08 in the healthy control group, which indicates a satisfactory functional level, with only occasional limitations. In the group undergoing chemotherapy, the average score was 6.25 ± 11.30, reflecting severe and frequent physical limitations in daily life, as reported by the participants themselves. Therefore, the pattern identified is not only coherent with the structure of the instrument, but also consistent with what is expected in the literature, which points to acute functional decline after chemotherapy.

    To avoid any interpretative doubts, we will insert an explanatory note in the legend of Table 1, as suggested: 

    "Note: SF-36 scores range from 0 to 100, with higher values indicating better health and functioning."

    Ware, J. E., Sherbourne, C. D. (1992). The MOS 36-item short-form health survey (SF-36): I. Conceptual framework and item selection. Medical Care, 30(6), 473–483.

  5. Comment: Please increase the font size and bars of Figure 2.
  6. Response: Thank you for your comment regarding the appearance of Figure 2. We would like to clarify that the current format of the graphs and the relative size of the letters were generated automatically by the software used to analyze and visualize the data. Unfortunately, due to the limitations of the tool, it was not possible to manually adjust the proportion of the graphs or resize the letters independently without compromising the integrity of the data presented. However, we have tried to balance readability and visual accuracy within these constraints and remain open to suggestions for future improvements.

Reviewer 4 Report

Comments and Suggestions for Authors

This article explores the impact of chemotherapy on eye movement in breast cancer patients. It compares 12 patients undergoing chemotherapy with 12 healthy controls, also considering quality-of-life factors. The study found altered eye movement patterns in patients, along with negative correlations between eye movement, chemotherapy cycles, functional capacity, and physical limitations. These results suggest chemotherapy may affect oculomotor function. However, the small sample size limits the study’s generalizability.

Material and Methods – Comments and Suggestions

  • Lacks a clear description of the quality of life parameters included in the study and how these were assessed.

Results - Comments and Suggestions

  • Line 154: “GHG” sigla is not defined anywhere.

  • Line 162: “CH” should maybe be CT.

Figures – Comments and Suggestions

  • All figures’ resolution should be improved.
  • Figure 2: The letters are too big compared to the charts, and the charts seem to be squished.  

Conclusion – Comments and Suggestions

  • The conclusions should be interpreted with caution due to the small sample size, which represents a major limitation of the study.

Author Response

This article explores the impact of chemotherapy on eye movement in breast cancer patients. It compares 12 patients undergoing chemotherapy with 12 healthy controls, also considering quality-of-life factors. The study found altered eye movement patterns in patients, along with negative correlations between eye movement, chemotherapy cycles, functional capacity, and physical limitations. These results suggest chemotherapy may affect oculomotor function. However, the small sample size limits the study’s generalizability.

Comment: Material and Methods – Comments and Suggestions

  • Lacks a clear description of the quality of life parameters included in the study and how these were assessed.

Response: We thank you for your pertinent comment, which has allowed us to improve the methodological clarity of the manuscript. In the revised version, we will include a more detailed description of the quality of life (QOL) parameters used in the research, as requested.
Thus, it has been included in procedures: 

“Quality of life was assessed using the Medical Outcomes Study 36-Item Short-Form Health Survey (SF-36). This is a self-report questionnaire made up of 36 items divided into eight domains: Functional Capacity (10 items), Physical Limitations (4 items), Pain (2 items), General Health Status (5 items), Vitality (4 items), Social Aspects (2 items), Emotional Limitations (3 items) and Mental Health (5 items). The scores for each domain are transformed into a scale from 0 to 100, where higher values indicate a better perception of health and quality of life.” 

Liem, Y.S., Bosch, J.L., Arends, L.R., M. H., Heijenbrok-Kal, Myriam Hunink, M.G. (2007).  Quality of life assessed with the Medical Outcomes Study Short Form 36-Item Health Survey of patients on renal replacement therapy: a systematic review and meta-analysis. Value Health, 10(5):390-7. doi: 10.1111/j.1524-4733.2007.00193.x.

Comment: Results - Comments and Suggestions

  • Line 154: “GHG” sigla is not defined anywhere.

  • Line 162: “CH” should maybe be CT.

Response: The correct term in the manuscript is CT (chemotherapy), not “CH”. The correction has been made in the updated version.

Comment: Figures – Comments and Suggestions

  • All figures’ resolution should be improved.
  • Figure 2: The letters are too big compared to the charts, and the charts seem to be squished.  

Response: Thank you for your comment regarding the appearance of Figure 2. We would like to clarify that the current format of the graphs and the relative size of the letters were generated automatically by the software used to analyze and visualize the data. Unfortunately, due to the limitations of the tool, it was not possible to manually adjust the proportion of the graphs or resize the letters independently without compromising the integrity of the data presented. However, we have tried to balance readability and visual accuracy within these constraints and remain open to suggestions for future improvements.

Comment: Conclusion – Comments and Suggestions

  • The conclusions should be interpreted with caution due to the small sample size, which represents a major limitation of the study.
  • Response: We thank you for your comment, which allows us to reinforce the cautious interpretation of the results. In the revised version of the manuscript, we will address this limitation more explicitly in the discussion section.

Round 2

Reviewer 1 Report

Comments and Suggestions for Authors

I realy like all responces. Thank You.

Author Response

I'm pleased to know that the responses were helpful. Should you require any further information, please do not hesitate to contact me.

Reviewer 4 Report

Comments and Suggestions for Authors

The authors have addressed all the questions raised during the review process. While I still believe that the quality of the images could be improved by using alternative software, if the current tools present limitations. Overall, I am satisfied with the revisions made and approve the current version of the manuscript for publication.

Author Response

Dear Reviewer,

Thank you for your thoughtful feedback and for approving the current version of the manuscript for publication.

I have made an effort to improve the quality of the images, within the limitations of the software currently available. 

Kind regards,